# Usefulness of Measuring Serum Amyloid A Concentration in Japanese Black Cattle in Clinical Practice

**DOI:** 10.3390/vetsci10080528

**Published:** 2023-08-17

**Authors:** Urara Shinya, Osamu Yamato, Yuka Iwamura, Tomohiro Kato, Yuhei Hamada, Oky Setyo Widodo, Masayasu Taniguchi, Mitsuhiro Takagi

**Affiliations:** 1Soo Veterinary Clinic, Kagoshima A.M.A.A., Soo 899-8212, Japan; shinya-u@nosai46.jp (U.S.); iwamura-y@nosai46.jp (Y.I.); kato-t@nosai46.jp (T.K.); hamada-y@nosai46.jp (Y.H.); 2Clinical Laboratory Training Center Eastern Laboratory, Kagoshima A.M.A.A., Soo 899-8212, Japan; 3Joint Faculty of Veterinary Medicine, Kagoshima University, Kagoshima 890-0065, Japan; osam@vet.kagoshima-u.ac.jp; 4Division of Animal Husbandry, Faculty of Veterinary Medicine, Universitas Airlangga, Surabaya 60115, Indonesia; oky.widodo@fkh.unair.ac.id; 5Joint Graduate School of Veterinary Sciences, Yamaguchi University, Yamaguchi 753-8515, Japan; masa0810@yamaguchi-u.ac.jp; 6Joint Faculty of Veterinary Medicine, Yamaguchi University, Yamaguchi 753-8515, Japan

**Keywords:** acute phase protein, diagnosis, inflammatory marker, Japanese Black, serum amyloid A

## Abstract

**Simple Summary:**

The results of this test suggest that serum amyloid A is a sensitive and reliable index of inflammation without disease specificity in Japanese Black cattle. Moreover, the serum amyloid A concentration can evaluate the prognosis and therapeutic effects in diseased cattle more objectively.

**Abstract:**

This study investigated the concentration of serum amyloid A (SAA), an acute phase protein, in Japanese Black cattle. Four practical trials were performed to evaluate the transition of SAA and sialic acid before and after dehorning, the relationship between the SAA concentration and other blood test parameters, the SAA dynamics in the diseased cattle, and the blood test results, including the SAA concentrations, of the two cases with a follow-up. The SAA concentration increased with dehorning but decreased 7 days after dehorning. The SAA concentration is positively correlated with the α-globulin, sialic acid, and fibrinogen concentrations and negatively correlated with the serum iron concentration. The SAA concentration in the deceased herd was significantly higher than that in the cured outcome herd. In addition, the SAA concentration in the cured group decreased significantly from the first test to retesting but increased significantly in the disuse group. Thus, SAA is a sensitive index of inflammation and a monitoring tool in Japanese Black cattle, and its measurement is considered useful in clinical practice.

## 1. Introduction

An acute phase protein (APP) is a specific protein whose blood concentration fluctuates within a short period when the body is subjected to harmful stimuli, such as infection or tissue damage. Since APPs can be used to determine the presence or absence and the degree of inflammation, therapeutic effects, and prognosis, they are considered highly useful indices in clinical practice. C-reactive protein (CRP), a typical APP in human medicine that reacts rapidly, particularly during inflammation, is commonly used to determine treatment strategy and prognosis [1,2]. CRP has been used as an index of inflammation in dogs [3] and horses [4] in veterinary medicine. However, it is not a suitable indicator of inflammation in cats and cows, as the increase in the CRP concentrations due to inflammatory stimulation is unclear [5,6]. Therefore, serum amyloid A (SAA), which is mainly synthesized in the liver and its synthesis is induced via inflammatory cytokines such as IL-1, IL-6, and TNF-α [7], has been used as an index of inflammation in cats in clinical practice [8,9]. Similar to CRP, SAA is an APP that responds rapidly to inflammation, and it has been used in humans [10,11] and horses in clinical practice [12,13]. Its concentration increases markedly during inflammation and decreases rapidly as the inflammation subsides. SAA concentration has been considered both as a potential indicator of disease and well-being in individual animals and as an indicator of herd health [14,15,16,17]. Sialic acid, an index of inflammation, is also measured at our facility. Since sialic acid is mainly bound to several APP in the blood, it has been used as an index of inflammation in humans [18,19] and animals [20,21,22] in clinical practice. However, sialic acid is inferior to SAA and CRP in terms of the agility and sharpness of the reaction during inflammation. In addition, in human medicine, the frequency of measuring sialic acid concentrations has decreased since the advent of the measurement of CRP concentrations, which is highly convenient. Although there have been several reports on the measurement of SAA concentrations in dairy cows [23,24] and Japanese Black suckling calves [25], no reports have been published on the SAA concentrations in Japanese Black breeding cattle. We recently reported a cut-off value of SAA (6.5 mg/L) for Japanese Black cattle for the first time [26]; therefore, as an additional verification, the usefulness of measuring the SAA concentrations in Japanese Black cattle using the cut-off value in clinical practice was evaluated.

## 2. Materials and Methods

This study was conducted in accordance with the regulations for the protection of experimental animals and the guidelines of Yamaguchi University, Yamaguchi, Japan (No.40, 1995, approved on 27 March 2017).

Blood analysis. Complete blood counts were assessed using an automatic blood cell counter (F-820; Sysmex, Tokyo, Japan). Serum biochemical analysis was performed (measured using Labospect 7180 autoanalyzer; Hitachi, Tokyo, Japan) to determine the following parameters: sialic acid in blood, glucose, free fatty acid, total cholesterol, total protein, albumin, albumin/globulin ratio, bilirubin, urea nitrogen, aspartate aminotransferase, γ-glutamyltransferase, calcium, magnesium, iron, and inorganic phosphorus. The SAA concentration was also measured using an automated biochemical analyzer (Labospect 7180 autoanalyzer or Pentra C200; HORIBA ABX SAS, Montpellier, France) with a special SAA reagent for animal serum or plasma (VET-SAA ‘Eiken’ reagent; Eiken Chemical Co., Ltd., Tokyo, Japan). The SAA concentration was calculated using a standard curve generated using a calibrator (VET-SAA calibrator set; Eiken Chemical Co., Ltd., Tokyo, Japan) according to our previous report (Shinya et al., 2022). Protein electrophoresis was conducted with a fully automated capillary electrophoresis system (Sebia, Tokyo, Japan), and the electrolytes were measured with an electrolyte analyzer (Techno Medica Co., Ltd., Yokohama, Japan). Fibrinogen was measured using a thermal precipitation refractometer method.

### 2.1. First Practical Trial: Transition of SAA and Sialic Acid Concentrations before and after Dehorning

Blood was collected from 19 11-month-old clinically healthy Japanese Black sterilized cows immediately before dehorning (day 0) and on days 1, 4, and 7 after dehorning. The serum was separated from the collected blood within 2 h of collection, and the SAA and sialic acid concentrations were measured. Friedman’s test was used to compare the concentrations of SAA and sialic acid on days 0, 1, 4, and 7.

### 2.2. Second Practical Trial: The Relationship between SAA and the Other Blood Test Parameters

The test group comprised 128 Japanese Black cattle that underwent haematological examinations due to illnesses between April 2021 and July 2022. The SAA concentrations and other blood parameters were measured at our facility, and the correlations between the SAA concentration and other parameters were analyzed using Spearman’s rank correlation coefficient. A list of the inspection parameters and the measurement methods of parameters other than SAA is mentioned in Appendix A.

### 2.3. Third Practical Trial: Investigation of SAA Dynamics in Diseased Cattle

Investigation of SAA dynamics in diseased cattle. During the second trial, 111 cows were classified into two groups: the cured (n = 72) and deceased groups (n = 39). The SAA concentrations measured in both groups were compared using the Mann–Whitney U test. The blood test parameters and methods were the same as those used in the second trial. Investigating the blood test results and clinical symptoms of cattle revealed a marked increase in the SAA concentration (≥100 mg/L) in the cured group and a relatively mild increase in the SAA concentration (≤20 mg/L) in the deceased group.

### 2.4. Fourth Practical Trial: Report of Blood Test Results, including the SAA Concentration, in Two Cases with Follow-Up

Report of the blood test results, including the SAA concentration, in two cases with a follow-up. We describe two cases in which changes in the SAA concentration were observed over time during a single treatment session. Case 1 involved a 4-month-old Japanese Black cattle that was cured, whereas Case 2 involved a 7-year-old Japanese Black cattle that was deceased. The SAA concentrations were measured twice, and blood tests were also performed twice at our institution during the medical examination in both cases. The blood test parameters and methods were the same as those used in the second trial.

## 3. Results

### 3.1. First Practical Trial: Transition of SAA and Sialic Acid before and after Dehorning

The SAA concentration increased from day 0 (8.4 ± 7.6 mg/L) to day 1 (13.2 ± 8.1 mg/L). Compared with that on day 1, the values on day 4 (6.9 ± 4.1 mg/L) and day 7 (5.3 ± 9.6 mg/L) were significantly lower (*p* < 0.01, Figure 1a). In contrast, no significant differences were observed between the sialic acid concentration on day 0 (48.9 ± 4.2 mg/dL), day 1 (49.8 ± 4.0 mg/dL), day 4 (50.5 ± 4.5 mg/dL), and day 7 (49.6 ± 4.5 mg/dL) (Figure 1b). The sialic acid concentration was below the sialic acid cut-off value (60 mg/dL) at our facility at all blood sampling points before and after dehorning.

### 3.2. Second Practical Trial: Relationship between the SAA Concentration and Other Blood Test Parameters

A significant positive correlation was observed between the SAA concentration and the α-globulin (r = 0.519, *p* < 0.01), sialic acid (r = 0.527, *p* < 0.01), and fibrinogen (r = 0.47, *p* < 0.01) concentrations. In contrast, a significant negative correlation was observed with the serum iron concentrations (r = −0.547, *p* < 0.01) (Figure 2). No correlations were observed between the other test items and the SAA concentration.

### 3.3. Third Practical Trial: Investigation of SAA Dynamics in the Diseased Cattle

Compared with the SAA concentration (41.5 ± 52.8 mg/L) of the cured group, the SAA concentration (84.4 ± 108.2 mg/L) of the deceased group was significantly higher (*p* ≤ 0.05) (Figure 3). Six of the seventy-two cattle in the cured group showed a marked increase in the SAA concentration (≥100 mg/L). Among them, three had bronchitis, one had enteritis, one had rumen retention, and one had birth canal lacerations. Table 1 presents the abnormal values observed in the blood test results of these six cattle. The α-globulin, fibrinogen, and serum iron concentrations were abnormal in all cattle. No abnormalities were observed in any other examination items other than those listed in Table 1, and no characteristic findings were observed in the examination request form or medical records. Eight of the thirty-nine cattle in the deceased group showed relatively mild increases in the SAA concentrations (≤20 mg/L). Among these eight cows, two had toxaemia during pregnancy, two had anaemia, one had cardiac disease, one had a urological disease, one had lochia stagnation, and one had ruminal indigestion. Only the α-globulin concentrations were abnormal in these cows (Table 2). Table 3 presents the results of the other parameters, test request forms, characteristic major abnormal values, and signs from the medical chart surveys. Three of these eight cattle showed signs of chronic inflammation (total protein level ≥ 8.0 mg/dL and γ-globulin level ≥ 3.2 g/dL).

### 3.4. Fourth Practical Trial: Blood Test Results, Including the SAA Concentrations, of the Two Cases with Follow-Up

#### 3.4.1. Case 1 Involved a 4-Month-Old Japanese Black Cattle

On the first day of illness, the patient had a body temperature of 37.9 °C, was unable to stand up, and had loose stools and ocular depression suspected to be due to dehydration. Blood tests were performed on days 2 and 5 (Table 4). On the second day of illness, the body temperature was 38.7 °C, and the patient was able to stand up with assistance; however, the patient had anorexia and no energy. On the fifth day of illness, the body temperature was 39.2 °C and although the patient was able to stand up, vitality and appetite were not recovered. The SAA concentration was 114.6 mg/L on the second day of illness; however, it decreased to 17.8 mg/L on the fifth day of illness. The patient was healed by day 11.

#### 3.4.2. Case 2 Involved a 7-Year-Old Japanese Black Cattle

On the first day of illness, the body temperature was 38.5 °C, and appetite was normal; however, difficulty in standing up was observed. Blood tests were performed on days 2 and 8 (Table 4). No significant clinical changes were observed between days 1 and 8. The SAA concentration was 65.7 mg/L on the second day of illness; however, it increased to 366.5 mg/L on the eighth day of illness.

## 4. Discussion

During the first trial, the SAA concentration increased on day 1 after dehorning; however, it decreased on days 4 and 7. The SAA cut-off value for Japanese Black steers is unknown. However, according to the cut-off value (6.5 mg/L) for Japanese Black cattle previously reported by the authors [26], the SAA concentration increased due to dehorning but recovered to a value within the reference range by day 7 after dehorning. The SAA concentration peaks on the second day after surgery and returns to normal within 7−14 days in horses [13]. In cats, the SAA concentration peaked 24−48 h after sterilization surgery and then recovered to a value within the reference range 4 days later [27]. The results of the present study are consistent with those of previous reports. It is unknown why the SAA concentration on day 0 (immediately before dehorning) was above the cut-off value of 6.5 mg/L; however, the SAA concentration in horses reportedly increases by approximately two-fold after intense training or competition [13]. Therefore, the effect of retention during dehorning was considered. In contrast, no change in the sialic acid concentration was observed immediately before dehorning and 7 days after dehorning. In addition, the concentrations at all blood collection points were below the cut-off value (60 g/dL) at our facility. Thus, the findings of this result clarified that the SAA concentration in Japanese Black steers is more sensitive to stimuli than sialic acid and reflects damage to the living body.

The results of the second trial showed a positive correlation between the SAA and the α-globulin, sialic acid, and fibrinogen concentrations and a negative correlation with the serum iron concentration. α-globulin, sialic acid, and fibrinogen are positive indicators of inflammation, whereas serum iron is considered a negative indicator of inflammation in cattle [28,29]. The findings of this study clarified that the concentration of SAA increased due to inflammation in Japanese Black cattle.

The results of the third trial showed that the SAA concentration in the deceased group was significantly higher than that in the cured group. Thus, diseased cows were less likely to heal if the SAA concentration was high at the time of diagnosis. However, six cattle from the healed groups (n = 72) showed marked elevations in the SAA concentrations (≥100 mg/L), and eight cattle in the deceased group (n = 39) showed relatively mild elevations in the SAA concentrations (≤20 mg/L) in this study. All six cattle in the healed group had abnormal α-globulin, fibrinogen, and serum iron concentrations. Two cattle had a higher free fatty acid concentration than the normal range; however, since the free fatty acid concentration increases when the cattle are in a state of negative energy balance, it is thought that the decrease in food intake due to the effects of inflammation was the cause. In addition, due to the absence of any other major findings, six cattle in the healing group were diagnosed with severe acute inflammation. In contrast, among the acute inflammatory parameters, only the α-globulin concentration was abnormal in the eight cattle from the deceased group. However, chronic inflammation was observed in three of the eight cattle. Thus, they had lesser acute inflammation, which was considered the predominant pathology, overlapping with the other findings and symptoms. In countries other than Japan, SAA is used as an index of inflammation in dairy and beef cattle [14,30,31,32]; however, it is difficult to diagnose respiratory diseases by measuring the APP concentrations only [33]. According to the results of this trial, SAA concentration reflects the presence and degree of inflammation non-specifically and can be used as a reference for prognostic criteria. However, it is considered difficult to diagnose diseased cattle only by measuring the SAA concentration.

The SAA concentration of Case 1 in the fourth trial was 114.6 mg/L on the second day of illness, which decreased to 17.8 mg/L on the fifth day of illness. Although the standard range of SAA concentration in Japanese Black cattle is unknown, it decreased significantly from the second to the fifth day of illness and then recovered. In contrast, the SAA concentration of case 2 was 65.7 mg/L on the second day of illness, which was significantly higher than the cut-off value (6.5 mg/L) previously reported by the authors [26]. The SAA concentration further increased to 366.5 mg/L on the eighth day of illness and was discontinued on the fourteenth day. Continuous measurement of the SAA concentration is effective in horses [13] and cats [9]. In addition, as previously reported by the authors, a diagnosis cannot be made based on the results of a single measurement of SAA in clinical settings; therefore, it is necessary to measure the SAA concentration multiple times during the course of the illness [26]. Based on the results of this study, the SAA concentration must decrease rapidly from the viewpoint of prognostic prediction in Japanese Black cattle, and it is considered beneficial for clinical veterinarians to record the changes in the SAA concentration during a single course of medical treatment.

## 5. Conclusions

The results of this study indicate that SAA is a sensitive and reliable index of inflammation without disease specificity in Japanese Black cattle, and it can be used to evaluate the prognosis and therapeutic effects in diseased cattle objectively. It is crucial to communicate the condition of the diseased cattle to the owners and farm workers objectively. The measurement of SAA concentration is useful for on-site veterinarians. In addition, evidence-based treatment is desired in the future from the perspective of drug therapy; therefore, SAA measurements are valuable in the field of veterinary medicine. However, measuring the concentrations of SAA alone is not sufficient to make a diagnosis; therefore, these measurements must be combined with other tests. Moreover, insurance coverage does not cover the cost of testing since SAA is currently not listed in the Livestock Mutual Aid points table. In the future, we aim to consider the cost and further investigate the SAA concentrations in Japanese Black cattle.

## Figures and Tables

**Figure 1 vetsci-10-00528-f001:**
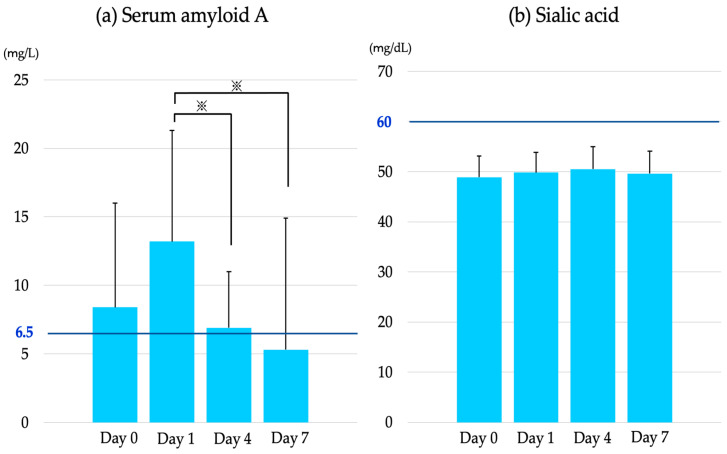
Concentrations of (**a**) serum amyloid A (cut-off value: 6.5 mg/L) and (**b**) sialic acid (cut-off value: 60 mg/dL) of examined cattle before and after dehorning. ^※^: *p* < 0.01.

**Figure 2 vetsci-10-00528-f002:**
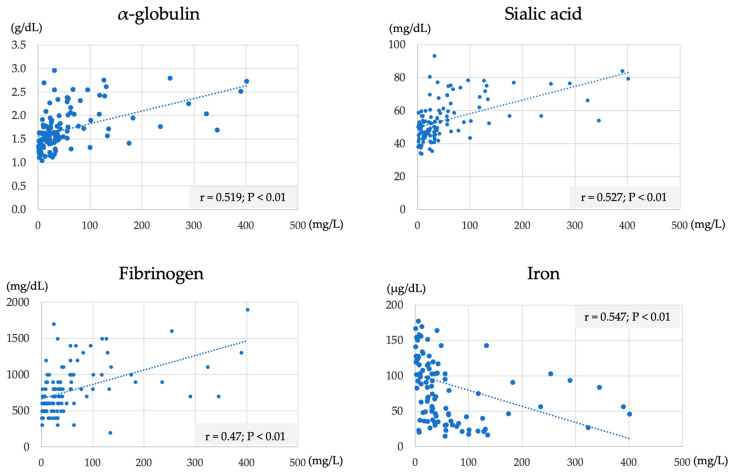
Representative significant correlation between the SAA concentration (X axis) and the α-globulin, sialic acid, fibrinogen, and iron concentrations.

**Figure 3 vetsci-10-00528-f003:**
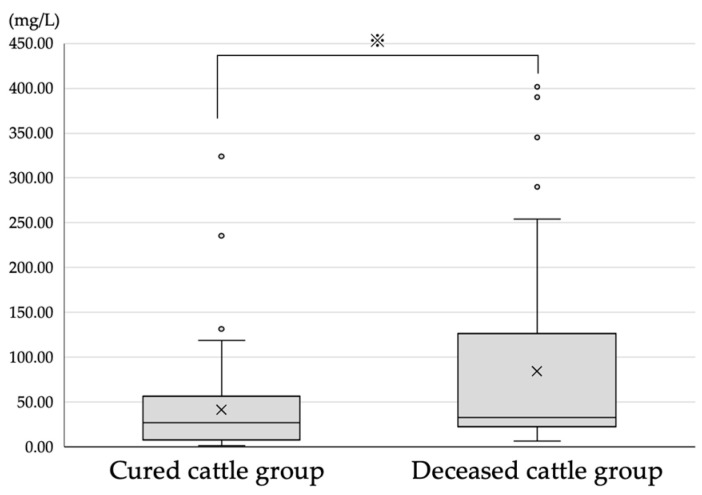
Serum amyloid A concentrations of cured and deceased groups. ^※^: *p* < 0.05.

**Table 1 vetsci-10-00528-t001:** Blood test results showing abnormal values in six cows that showed a marked increase in the SAA concentration (≥100 mg/L) in the cured group.

Symptom of the Cattle	SAA (mg/L)	WBC	Lymphocyte (%)	α-Globulin (g/dL)	Fibrinogen (mg/dL)	Sialic Acid (mg/dL)	Fe (μg/dL)	TP (g/dL)	ALB (g/dL)	A/G	FFA (μEq/dL)
Bronchitis (1)	324.1	8900	25 ↓	2.03 ↑	1100 ↑	66.1 ↑	26.4 ↓	7.17	3.51	0.96	962.8 ↑
Rumen retention	235.3	11,400 ↑	44	1.76 ↑	900 ↑	56.7	55.8 ↓	6.56	3.01	0.85	302.9
Birth canal laceration	136.1	4200	47	1.71 ↑	1100 ↑	52.2	15.9 ↓	7.4	3.79	1.05	770.8
Bronchitis (2)	131.7	11,500 ↑	28 ↓	2.61 ↑	800 ↑	75.1 ↑	24.3 ↓	8.81 ↑	2.28 ↓	0.35 ↓	218.5
Bronchitis (3)	118.7	8800	25 ↓	2.43 ↑	1500 ↑	68.2 ↑	74.8 ↓	7.83	2.94 ↓	0.60 ↓	196.4
Enteritis	118	5600	35	2.02 ↑	1000 ↑	61.9 ↑	21.6 ↓	7.92	3.08	0.64 ↓	987.5 ↑
Outliers (%)	100	33	50	100	100	66	100	16.6	33	50	33

↑: Higher than the standard value at this facility; ↓: Lower than the standard value at this facility.

**Table 2 vetsci-10-00528-t002:** Blood test results for inflammation in eight cows with relatively mild SAA concentrations (≤20 mg/L) in the deceased group.

	SAA (mg/L)	WBC	Lymphocyte (%)	α-Globulin (g/dL)	Fibrinogen (mg/dL)	Sialic Acid (mg/dL)	Fe (μg/dL)	TP (g/dL)	ALB (g/dL)	A/G
Toxemia of pregnancy (1)	6.7	2800	48	1.35 ↑	700	49.9	126.6	9.31 ↑	3.39	0.57 ↓
Toxemia of pregnancy (2)	9.7	9400	53	1.47 ↑	500	54	155.3	7.01	3.26	0.73
Anemia (1)	7.8	7400	43	1.35 ↑	600	56.8	19.9 ↓	3.96 ↓	1.53 ↓	0.63 ↓
Anemia (2)	8.3	9800	55	1.62 ↑	800 ↑	50.7	344.8	8.15 ↑	3.03	0.59 ↓
Cardiac disease	8.9	10,100 ↑	42	1.91 ↑	1200 ↑	56.6	37.1 ↓	7.18	2.53 ↓	0.54 ↓
Urological disease	10.1	15,900 ↑	54	1.79 ↑	500	56.8	97.9	7.67	3.65	0.91
Lochia stagnation	11.3	20,200 ↑	28 ↓	1.75 ↑	900 ↑	53	155.7	7.2	3.15	0.78
Ruminal indigestion	17.1	11,000 ↑	21 ↓	1.44 ↑	400	47.1	35.8 ↓	6.44	2.59 ↓	0.67
Outliers (%)	100	50	25	100	37	0	37	37	37	50

↑: Higher than the standard value at this facility; ↓: Lower than the standard value at this facility.

**Table 3 vetsci-10-00528-t003:** Characteristic findings of the eight cows with a relatively mild elevation of SAA concentration (≤20 mg/L) in the deceased group.

	SAA (mg/L)	Chronic Inflammation (TP ≥ 8.0 mg/dL; γ-Globulin ≥ 3.2 g/dL)	Main Abnormal Parameter or Clinical Symptoms	Other Findings
Toxemia of pregnancy (1)	6.7	√	FFA ≥ 1000 μEq/dL; 3HB ≥ 1000 μmol/L	After the examination, the calf with bicephalic malformation was delivered by caesarean section
Toxemia of pregnancy (2)	9.7	×	FFA ≥ 1000 μEq/dL; 3HB ≥ 1000 μmol/L	Acute bloating developed when the patient was caught in a fence and the patient could not stand up
Anemia (1)	7.8	×	Ht = 9.0%; RBC = 189 × 10^4^	−
Anemia (2)	8.3	√	Ht = 9.0%; RBC = 173 × 10^4^	−
Cardiac disease	8.9	×	Heart murmur; Transvenous distention	−
Urological disease	10.1	×	BUN = 72.4 mg/dL; AST = 124.8 IU/L	−
Lochia stagnation	11.3	√	−	Post-abortion lochia stagnation
Ruminal indigestion	17.1	×	Ht = 48.0%; Ca = 6.8 mg/dL	−

**Table 4 vetsci-10-00528-t004:** Blood test results of Japanese Black cattle who were cured and disused.

	Case 1		Case 2	
	2nd Sick Day	5th Sick Day	2nd Sick Day	8th Sick Day
TP (g/dL)	7.3	6	7.5	3.7 ↓
ALB (g/dL)	4.1 ↑	3.4	3.8 ↑	1.3 ↓
A/G	1.24	1.34	1.04	0.55 ↓
AST (IU/L)	107.9 ↑	333.8 ↑	330.4 ↑	199.5 ↑
GGT (IU/L)	28.6	19.6	11.5	6.3
T-Chol (mg/dL)	109.6	70.6	158.4	218.9 ↑
Glu (mg/dL)	391.9 ↑	100.5	65.9 ↑	55.8
FFA (μEq/L)	Over the limit	909.2 ↑	808.2 ↑	531.0 ↑
BUN (mg/dL)	88.5 ↑	7.9	8.4	85.5 ↑
Bill (mg/dL)	1.8 ↑	0.2	0.4 ↑	0.2
Ca (mg/dL)	9.1 ↓	9.9	10.2 ↓	5.6 ↓
IP (mg/dL)	7.7	4.2	7.2	7.3
Mg (mg/dL)	3.3	1.8	1.7	2.2
Fe (μg/dL)	43	73.1	135	78.0 ↑
Sialic acid (mg/dL)	121.1 ↑	86.4 ↑	43.4	38.4
SAA (mg/L) *	114.6 ↑	17.8 ↑	65.7 ↑	366.5 ↑
Na (mmol/L)	116.6 ↓	129.1 ↓	144.4	128.5 ↓
K (mmol/L)	1.64 ↓	3.23	4.25	5.11
Cl (mmol/L)	50.0 ↓	103.5	101.3	93.6 ↓
RBC (10^4^)	1609 ↑	1278	688	709
WBC	41,200 ↑	16,800 ↑	5900	9900
Lymphocyte (%)	10 ↓	26 ↓	45	42
α-globulin (g/dL)	2.05 ↑	1.64 ↑	1.38 ↑	1.41 ↑
β-globulin (g/dL)	1.33	0.55	0.39	0.21
γ-globulin	1.09	1.2	2.33	1.04
Fibrinogen (mg/dL)	800 ↑	700	900 ↑	1500 ↑

↑: Higher than the standard value at this facility; ↓: Lower than the standard value at this facility. *: SAA concentration is determined by referring to the cut-off value of Japanese Black breeding cattle.

## Data Availability

The original contributions presented in the study are included in the article/Appendix A. Further inquiries can be directed to the corresponding author.

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
