# Peer review of "Usefulness of Measuring Serum Amyloid A Concentration in Japanese Black Cattle in Clinical Practice"

_vetsci, 2023, doi:10.3390/vetsci10080528_

Round 1
Reviewer 1 Report
Serum amyloid A has already been applied for the clinical test to monitor tissue injuries and disorder inflammation, like in horses and humans. The author applied this index to monitor Japanese Black cattle and completed four practical trials. The results showed this index is more sensitive than the current parameters in the blood test, which indicated that Serum amyloid A may be more efficient and accurate in the clinical diagnosis. In the future, if the author can find a way to monitor this index more conveniently and simpler would be better.
Author Response
Veterinary Sciences
Manuscript ID: vetsci-2549971
Title:
Usefulness of measuring serum amyloid A concentration in Japanese Black cattle in clinical practice
We have revised the manuscript in accordance with the suggestions of Reviewer 1 as follows. Both the revised sections of the manuscript and the responses to the Reviewer below are marked in red.
Comments and Suggestions
Serum amyloid A has already been applied for the clinical test to monitor tissue injuries and disorder inflammation, like in horses and humans. The author applied this index to monitor Japanese Black cattle and completed four practical trials. The results showed this index is more sensitive than the current parameters in the blood test, which indicated that Serum amyloid A may be more efficient and accurate in the clinical diagnosis. In the future, if the author can find a way to monitor this index more conveniently and simpler would be better.
Thank you very much for your insightful comments. As stated in the comments we received, we will continue our clinical research in the future and strive to accumulate a simpler and more useful database in the clinical field of cattle production.

Reviewer 2 Report
Well designed three part research approach to address possible improvements in disease progression in Japanese Black Cattle.
For readers additional background on SAA in cattle and the usefulness in the detection of inflammation may be helpful. Why is inflammation a disadvantage specifically in cattle. Reduced economic return and possible spread of disease to members of the herd.
Please consider shortening some of your tables and move tables that have little value within the text to the supplement. Most of your graphs are very helpful.
Author Response
Veterinary Sciences
Manuscript ID: vetsci-2549971
Title:
Usefulness of measuring serum amyloid A concentration in Japanese Black cattle in clinical practice
We have revised the manuscript in accordance with the suggestions of Reviewer 2 as follows. The responses to the Reviewer below are marked in blue.
Comments and Suggestions
a well-designed three-part research approach to address possible improvements in disease progression in Japanese Black Cattle.
For readers, additional background on SAA in cattle and its usefulness in the detection of inflammation may be helpful. Why is inflammation a disadvantage, specifically in cattle? Reduced economic return and the possible spread of disease to members of the herd.
Please consider shortening some of your tables and moving tables that have little value within the text to the supplement. Most of your graphs are very helpful.
Thank you very much for your insightful comments. Following your comment, in the "Introduction" part, we have added the sentences on the background of SAA and its usefulness in the detection of inflammation in cattle clinical practice, Lines 49–50 and Lines 56–58. Additionally, we added four references: lines 318–319, line 334, lines 335–336, and lines 347–348.
Additionally, to reduce the number of tables, we moved Table 1 to Supplement and made it Table S1, mentioned in Lines 106–107. And also, we modify the Table numbers (Table 2, Table 3, Table 4, and Table 5) to (Table 1, Table 2, Table 3, and Table 4).
